# Multi-Objective Control Strategy for Switched Reluctance Generators in Small-Scale Wind Power Generations

Linqiang Wang [1] , Cheng Liu [2], Zongwen Jiang [1], Weiren Xiao [1], Shuaiwei Ren [1], Jiaxin Ding [1] and Qing Wang [1,*]

1   School of Information Engineering, Nanchang University, Nanchang 330031, China
2   School of Advanced Manufacturing, Nanchang University, Nanchang 330031, China; liucheng08@ncu.edu.cn
*   Correspondence: wangq@ncu.edu.cn

**Abstract:** Switched reluctance generators (SRGs) are widely used in wind power generation. However, due to the natural tendency of SRG, there are always nonnegligible conflicts to achieve high efficiency and low output voltage ripple at the same time. This creates difficulties for the high-performance of SRG. Thus, a multi-objective optimization control strategy is proposed in this paper to improve the static performance of SRG. The proposed control strategy contains following steps. First, in order to gain the maximum output power range at different rotor speeds, the turn-off angle is optimized off-line by simulated annealing algorithm (SAA). The optimized results are fitted as a function of rotor speed for on-line regulating; then, a closed-loop controller is built, and the reference current is regulated according to the difference between actual output power and required output power. Second, a multi-objective function is constructed as the evaluation result of SRG performance, which takes system efficiency, output voltage ripple and power converter loss into consideration. In the end, the turn-on angle is tuned by SAA according to the real-time multi-objective evaluation result. The proposed control strategy can be flexibly applied to SRGs with different structures and avoids the disadvantage of single-objective optimization. The simulation and experiments results show that the overall performance of SRG is improved.

**Keywords:** switched reluctance generator; system efficiency; output voltage ripple; power converter loss; multi-objective control

## 1. Introduction

The switched reluctance machine (SRM) is widely used in wind power generation, electric vehicles and other fields because of its simple structure, high reliability, wide speed range and tolerance in harsh environments [1–4]. However, due to its doubly-salient structure, the performance of SRM is sensitive to phase current trajectory. SRM presents some known drawbacks, such as nonlinear inductance, high current and torque ripple, and acoustic noise, have hindered the mass adoption of such machines [5–7]. Therefore, many scholars have conducted in-depth research on performance optimization for SRM.

In published research papers, different control methods were suggested for SRGs to achieve high operating performance. In [8], direct power control (DPC) methods were suggested to capture more wind generation. But loss is not considered during the process, which cannot reflect actual system efficiency. In [9], a single pulse control with adopting freewheeling technique was designed for SRGs to improve indicators such as power generation, system efficiency, torque ripple and DC current ripple. The method was shown to improve the torque ripple and DC current ripple, however cannot enhance the output power and efficiency. A novel switching control strategy was proposed in [10], where three switching states of the converter were reassigned within a working cycle to improve the efficiency of SRG. But the switching control strategy can only operate at low-speed operation. A double-loop compensation voltage control method is proposed in [11], which can not only increase the power generation, but also effectively reduce the steady-state error, and has a good robustness.

The SRG's high performance operation can be achieved by adjusting turn-on and turn-off angles, which has been proven. In [12], the genetic algorithm was suggested for turn-on angle optimization to achieve maximum voltage output. But the strategy strongly relies on an accurate mathematic model. In [13], the finite element model was used for turn-on angle optimization to maximize its efficiency. However, it is also hard to build a mathematic model for power converter loss and stray loss in the system. Moreover, in [14], the fuzzy logic algorithm was suggested for turn-on and turn-off angles regulation. The method can restrain the voltage ripple and strengthen system's static performance well. Reported algorithms need either giant data storage or a simplified system model, which brings difficulties to popularizing multi-objective control in the SRG system [15].

Since single-objective optimization might reduce the performance of other indicators, multi-objective optimization was suggested to raise the overall system performance [16–20]. In [21], torque ripple, output power and current ripple were considered for turn-on angle optimizations in SRG, the multi-objective function for optimization with weight coefficients was proposed. But this is an off-line optimization method. On-line optimization is more convenient and accurate for multi-objective optimization [22,23].

Online estimation of system parameters is important for optimization process. For generators, system efficiency is the ratio of output power to input power. The input mechanical power needs to be calculated by calculating the torque. Typically, torque is calculated according to the derivative of magnetic co-energy. However, the calculation process is very complicated [24]. In [25], torque was fitted by a function of rotor position and phase current, which needs off-line preparation to determine the characteristics. In [26], the torque was estimated with the local linearization of energy conversion loop and the proposed method shows good quickness and accuracy. But the unsaturated inductance of the machine is still needed. For power converter loss, power loss models that can calculate and predict power converter loss were developed [27,28]. However, calculations involving switching loss is complex.

This paper proposes a multi-objective control strategy for SRG on-line performance optimization. A multi-objective function with weight coefficients is proposed, and system efficiency, voltage ripple and power converter loss are considered. First, the turn-off angle is optimized off-line by SAA to achieve maximum output power. Then, the reference current is regulated by the power controller to bring the actual output power to the target value. At last, the turn-on angle is optimized on-line according to the multi-objective evaluation result by SAA. The key objectives of this proposed multi-objective control strategy are as follows:

- The output power can meet the users' requirement and remains stable under the dynamic changes of wind speed. The entire optimization system can keep up with dynamic changes in wind speed.
- A multi-objective function including system efficiency, voltage ripple and power converter loss is constructed, which avoids the disadvantage of single-objective optimization. The overall performance of SRG is improved by online adjustment of the turn-on angle.
- The proposed strategy can be flexibly applied to SRG with different structures and does not require complex mathematical models. The optimization system is stable and easy to implement, which is conducive to the promotion of SRG in wind power generation.

The rest of this paper is organized as follows. Section 2 presents operating principles of SRG, including characteristics of power generation, characteristics of efficiency and characteristics of output voltage ripple. Section 3 presents the proposed multi-objective control strategy. The simulation analysis of the proposed system is presented in Section 4. Section 5 presents the experimental results, and finally, the conclusions are presented in Section 6.

## 2. Operating Principles of SRG

### 2.1. Characteristics of Power Generation

Figure 1 shows the structure of the SRG system. For small-scale wind power generations, wind turbine is employed as the prime mover. The output power of SRG system can be expressed as

$$P = U_{DC} \times I_{DC} \tag{1}$$

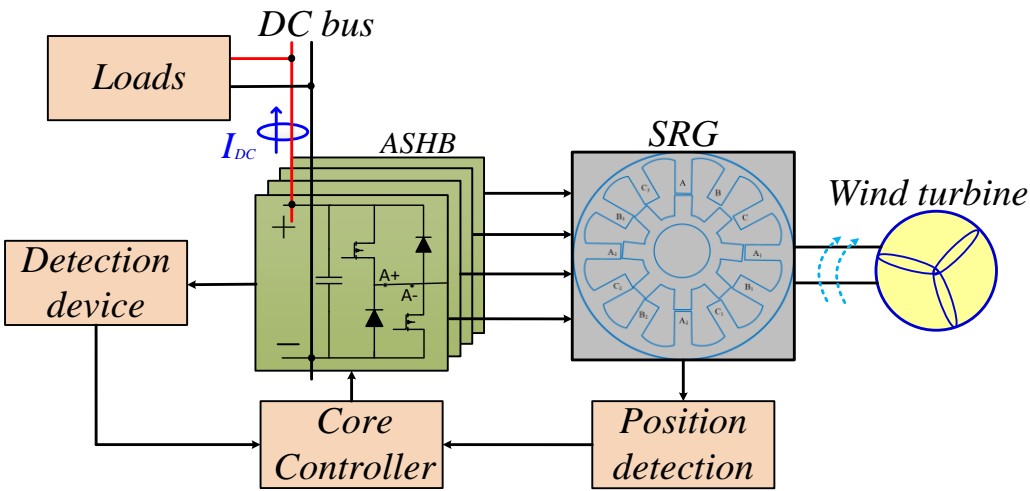

**Figure 1.** Switched reluctance generation system.

Figure 2 shows energy conversion loops under different turn-on and turn-off angles. For generator operations, phase windings absorb mechanical energy from the prime mover. In one stroke, the absorbed mechanical energy is proportional to the area enclosed by the flux-linkage trajectory.

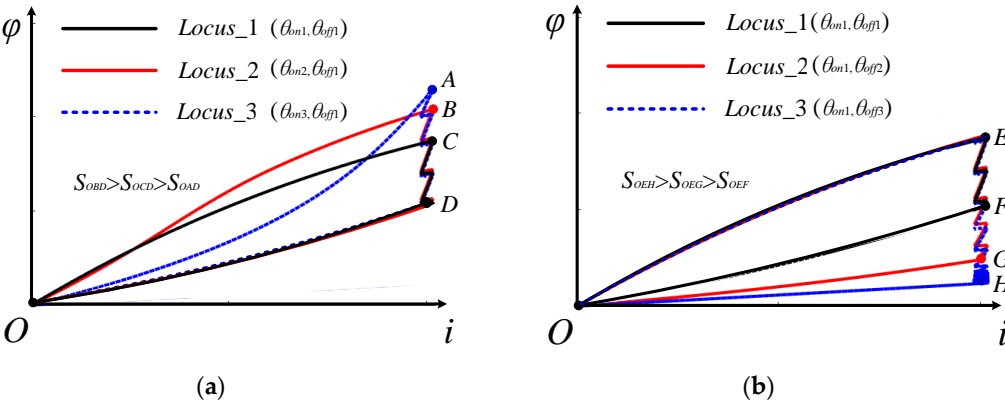

(**a**)                 (**b**)

**Figure 2.** Flux-linkage trajectories under different turn-on and turn-off angles:(**a**) The turn-off angle is fixed and the turn-on angle is changed; (**b**) The turn-on angle is fixed and the turn-off angle is changed.

As shown in Figure 2a, where the turn-off angle is fixed and $\theta_{on1} > \theta_{on2} > \theta_{on3}$. $S_{OCD}$, $S_{OBD}$ and $S_{OAD}$ can be approximated as the area surrounded by three flux-link trajectories, respectively. Reducing the conduction angle from $\theta_{on1}$ to $\theta_{on2}$, the absorbed mechanical energy is increased because the inductance increasing region can be better utilized. Decreasing the turn-on angle from $\theta_{on2}$ to $\theta_{on3}$, absorbed mechanical energy is decreased because the phase current is built in the inductance increasing region. As shown in Figure 2b, where the turn-on angle is fixed and $\theta_{off1} < \theta_{off2} < \theta_{off3}$. $S_{OEF}$, $S_{OEG}$ and $S_{OEH}$ can be approximated as the area surrounded by three flux-link trajectories, respectively. Increasing the turn-off angle, absorbed mechanical energy is increased. Thus, the maximum output power under constant phase current constraint is sensitive to both turn-on and turn-off angles.

### 2.2. Characteristics of Efficiency

Due to the system loss, the relationship between output electric power $P$ and absorbed mechanical power $P_m$ can be expressed as

$$P = \eta P_m \tag{2}$$

According to the electromechanical energy conversion theory, the mechanical energy $\Delta W$ absorbed by the phase winding in one stroke can be expressed as

$$\Delta W = m \int_{\varphi(\theta_{on})}^{\varphi(\theta_{ext})} i_k d\varphi = m \int_{t(\theta_{on})}^{t(\theta_{ext})} i_k(U_k - i_k R_{ESR})dt \tag{3}$$

Thus, absorbed mechanical power $P_m$ can be obtained as

$$P_m = \frac{\omega N_r}{2\pi} \Delta W \tag{4}$$

Since the equivalent resistance changes with the operating conditions, $R_{ESR}$ needs to be refreshed in real time. Considering the flux-linkage of each phase winding of SRG can be expressed as

$$\varphi_{ph} = \int (U_k - i_k R_{ESR})dt + \varphi_0 \tag{5}$$

At the turn-on position, both the initial flux $\varphi_0$ and phase current are 0, so

$$0 = \int_{\theta_{on}}^{\theta_{ext}} i_k(U_k - i_k R_{ESR})d\theta \tag{6}$$

The equivalent internal resistance $R_{ESR}$ can be thus obtained by

$$R_{ESR} = \frac{\int_{\theta_{on}}^{\theta_{ext}} U_k d\theta}{\int_{\theta_{on}}^{\theta_{ext}} i_k d\theta} \tag{7}$$

With (1)~(7), the system efficiency can be obtained by solving

$$\eta = \frac{P}{P_m} = 2\pi \frac{U_{DC} \times I_{DC}}{m N_r \Delta W} \tag{8}$$

Besides, power loss in SRG system includes loss in the machine and loss in the power converter. Since failure in power converter is the most common fault for SRG, loss in the power converter should be reduced to rise the system reliability. Loss in the power converter can be observed by

$$\Delta P_{PC} = P_{pc-in} - P_{pc-out} = average(\sum_{k=A,B,C\text{and } D} (U_k \times i_k)) - U_{DC} \times I_{DC} \tag{9}$$

It should be noted that high system efficiency leads to high utilization of wind power; low power converter loss leads to less temperature raise and high reliability in the converter. Thus, both $\eta$ and $\Delta P$ should be considered during the optimization.

### 2.3. Characteristics of Output Voltage Ripple

The typical relationship between phase current and output voltage is shown in Figure 3. As shown in the figure, the output voltage ripple can be classified as high-frequency voltage ripple, which is caused by current chopping and shown as $\Delta U_1$, and low-frequency voltage ripple, which is caused by commutation and shown as $\Delta U_2$.

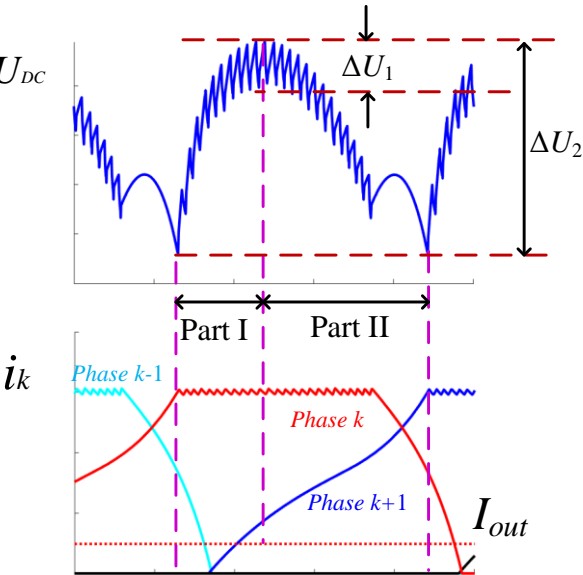

**Figure 3.** Voltage ripple and phase current waveform.

It is clear that the $\Delta U_1$ is sensitive to the chopping frequency and is far smaller than the low-frequency voltage ripple $\Delta U_2$. The change of $\Delta U_2$ can be divided into two parts as following.

(i) Part I: $i_g(k) > i_e(k+1) + I_{out}$. In this part, the outgoing phase (phase $k$ and phase $k-1$) operates in the power delivering mode and the total delivered current is recorded as $i_g(k)$; the incoming phase (phase $k+1$) operates in the excitation mode and excitation current is recorded as $i_e(k+1)$. The output voltage rises and the change of rate of output voltage is

$$\frac{dU_{DC}(t)}{dt} = \frac{1}{C}(i_g(k) - i_e(k+1) - I_{DC}) \tag{10}$$

When $i_g(k) = i_e(k+1) + I_{DC}$, the output voltage reaches the maximum value $U_{max}$.

(ii) Part II: $i_g(k) < i_e(k+1) + I_{out}$. In this part, the output voltage falls and the change of rate of output voltage can be also calculated according to (10). When the incoming phase ends the excitation operation, the output voltage reaches the minimum value $U_{min}$. Considering phase current chopping, $i_g(k)$ is expressed as

$$i_g(k) = i(k-1) + i(k) \times d \tag{11}$$

Thus, $\Delta U_2$ can be employed to reflect the output voltage ripple and is also sensitive to turn-on angles.

According to the analysis above, maximum output power range, system efficiency, power converter loss and output voltage ripple are important to evaluate the performance of SRG, all of which are sensitive to turn-on angles. Therefore, a multi-objective control strategy is proposed in the next section.

## 3. Multi-Objective Control

As generally recognized, turn-on angle, turn-off angle and reference current are major factors for performance optimization. Thus, the proposed multi-objective control strategy includes three steps. First, the turn-off angle is optimized off-line to guarantee the maximum output power at concerned rotor speeds. The optimized results are fitted as a function of rotor speed for on-line regulating. Then, a closed-loop controller is built, and the reference current is regulated according to the difference between required output power and actual output power. In the end, the turn-on angle is tuned on-line for multi-objective optimization.

For easy description, the multi-objective control strategy will be proposed on a given 8/6 SRG for case study and the structure of given SRG is shown as Figure 4.

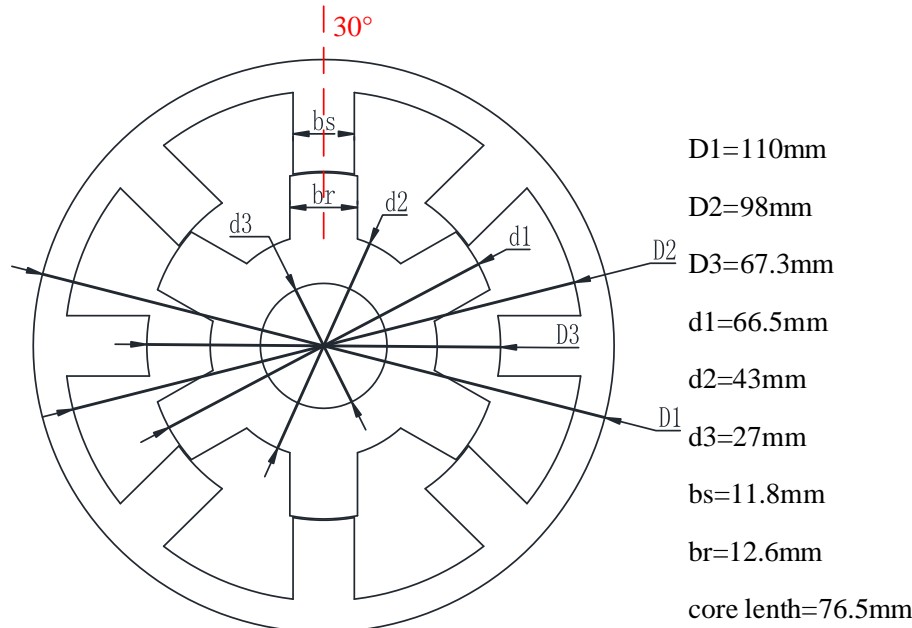

Figure 4. Structure of the given SRG for case study.

### 3.1. Off-Line Turn-Off Angle Optimization

In order to make full use of the inductance decreasing region, the initial turn-on angel is fixed at the position where the rotor teeth and stator teeth about to align (i.e., $\theta_{on\text{-}initial} = 20°$). Then, the turn-off angle is optimized by SAA to achieve maximum output power at different rotor speeds. The flow chart of SAA is shown as Figure 5.

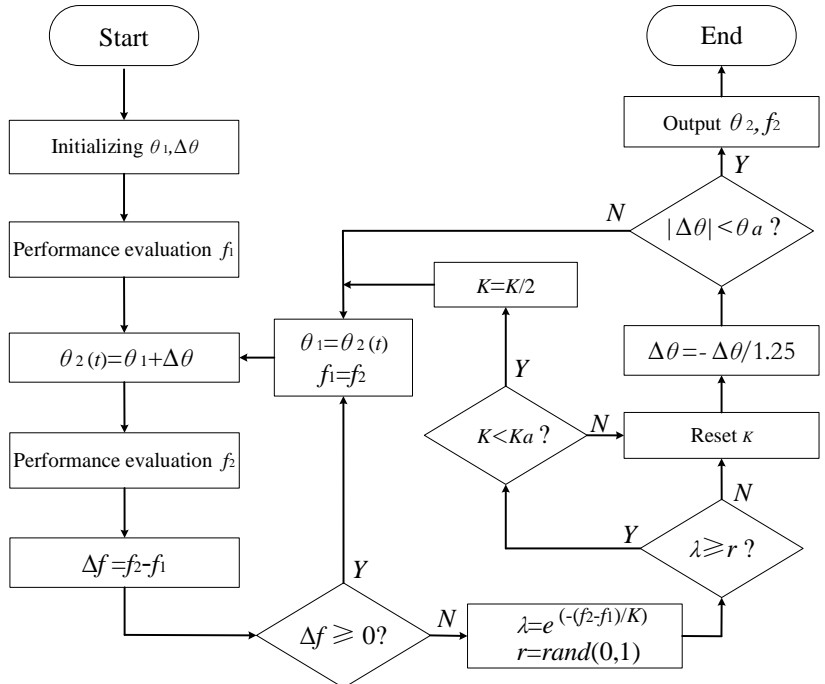

Figure 5. Flow chart of SAA.

As shown in the figure, SAA is similar to the climbing searching algorithm but can avoid local optimum. For turn-off optimization, the optimizer evaluates the system perfor-

mance according to the output power (i.e., *f* stands for the output power in Figure 5). If the latest performance is improved (i.e., $\Delta f > 0$), the optimized angle is refreshed. Otherwise, the optimized angle is refreshed according to Metropolis rules and the disturbance is changed (i.e., $\Delta\theta = -\Delta\theta/1.25$). During the optimization process, the initial turn-off angle is set as $\theta_1 = 42°$; initial disturbance is set as $\Delta\theta = 1.44°$; coefficient *K* and $K_a$ are set as $K = 0.01$ and $K_a = 0.001$, respectively; the minimum disturbance $\theta_a$ is set as $\theta_a = 0.18°$.

In Figure 6, optimized turn-off angles under different rotor speeds are shown as green spots in the figure. The red dashed line in the figure shows the curve fitting results according to (12), by which the turn-off angle can be regulated according to real-time rotor speed *n* on-line.

$$\theta_{off} = 49.85 - 0.4815\cos(0.007212 \times n) + 0.1675\sin(0.007212 \times n) \tag{12}$$

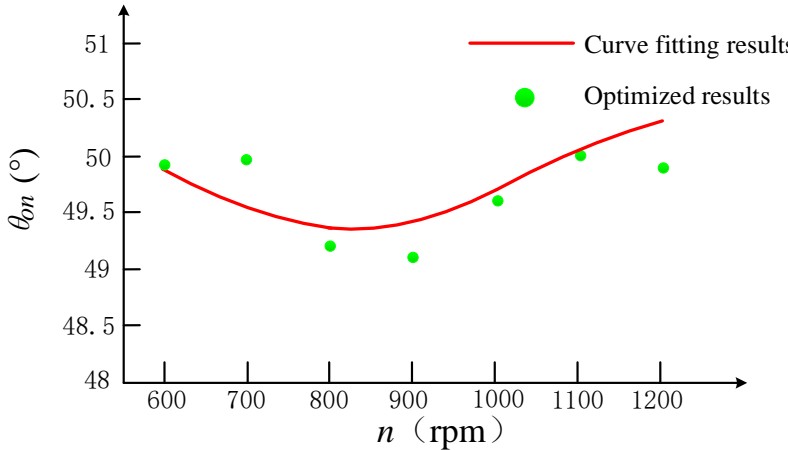

**Figure 6.** Results of turn-off angle optimization.

### 3.2. Multi-Objective Control and On-Line Turn-On Angle Tuning

As mentioned in Section 2, system efficiency, voltage ripple and power converter loss are considered for the on-line optimization by tuning the turn-on angle. A multi-objective function is built as following.

$$f(x) = \lambda_1 \frac{\eta}{\eta_{\max}} + \lambda_2 \frac{1/\Delta U}{(1/\Delta U)_{\max}} + \lambda_3 \frac{1/\Delta P_{PC}}{(1/\Delta P_{PC})_{\max}} \tag{13}$$

$$\lambda_1 + \lambda_2 + \lambda_3 = 1 \tag{14}$$

where, $\lambda_1$, $\lambda_2$ and $\lambda_3$ represent the weight coefficients of system efficiency, voltage ripple and power converter loss respectively. The weight coefficient of each optimization target should be reasonably distributed according to the focus of the optimization goal and the needs of the operating environment. In this paper, weight coefficients are set as $\lambda_1 = 0.5$, $\lambda_2 = 0.3$, $\lambda_3 = 0.2$.

The structure of the proposed multi-objective control strategy is shown as Figure 7, the output power closed-loop controller in the red box and the multi-objective optimization controller in the blue box. In the proposed control strategy, the control process is divided into the following steps.

(i) According to the difference between the required output power $P^*$ and actual output voltage $P(t)$, the reference current $i^*$ is regulated by the PI regulator.

(ii) Throughout the optimization process, the turn-off angle $\theta_{off}$ is calculated according to the real-time rotor speed in regulator2 by (12).

(iii) After the actual output power $P(t)$ reaches the target value $P^*$, the controller evaluates the system performance according to (13) and (14).

(iv) Finally, according to the evaluation in iii, the regulator1 tunes the turn-on angle on-line by SAA and the flow chart is shown as Figure 5.

During the turn-on angle tuning on-line, the proposed control strategy set the output power as the top control objective, which meets the users' first requirement. For turn-on optimization, the optimizer evaluates the system performance according to the multi-objective function (i.e., $f$ stands for the multi-objective function in Figure 5).

For generators, the change in operating point mainly includes the change of given output power and motor speed. For output power, the system adopts power closed-loop control, so the current operating power point can be derived from the required output power. Considering the wind speed is inconstant, the maximum output power variation range is set. If the disturbance of output power caused by the change in wind speed exceeds the range, then the turn-on angle is re-optimized. The maximum output power variation range can be reasonably distributed according to the needs of the operating environment. In this paper, the range is set to 5 W.

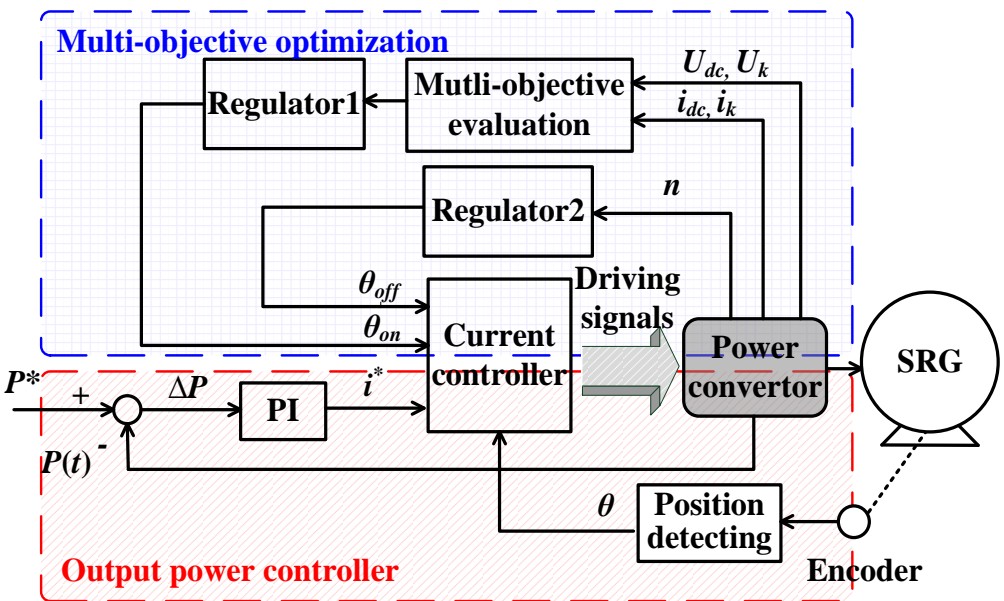

**Figure 7.** Structure of multi-objective controller.

## 4. Simulation Analysis

In order to verify the feasibility of proposed control strategy, the given SRG, the structure of which is shown as Figure 4, is employed for proof-of-concept.

Figure 8 shows the relationship between output power and turn-off angle at different rotor speeds, where the turn-on angle is fixed at $20°$, the phase current constraint is set as 10 A and the range of turn-off angle is set as $44–54°$. Section 2 presents that the maximum output power under constant phase current constraint is sensitive to both turn-on and turn-off angles. As shown in the figure, when the rotor speed is fixed, increasing the turn-off angle, absorbed mechanical energy is increased, so the output power is increased. When the turn-off angle enters the unaligned position, the effective generation range (the inductance drop region) decreases, resulting in the output power being reduced. So, there is an optimal turn-off angle to maximize the output power.

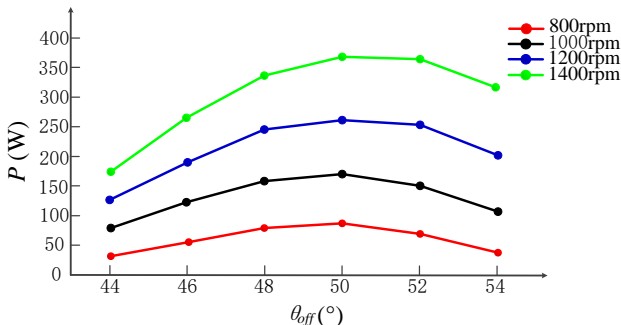

**Figure 8.** The relationship between turn-off angle, rotor speed and power generation.

Figure 9 shows the relationship between concerned control objectives and turn-on angle at different output power, where the turn-off angle is regulated according to (12) and the rotor speed is fixed at 800 rpm. As shown in Figure 9a, system efficiency can be improved by increasing the turn-on angle. However, the increase in the turn-on angle leads to an increase in the reference current and copper consumption. So, the closer turn-on angle to the align position, the slower system efficiency will be increased. Although the simulation cannot reflect accurate system loss, the trend of system efficiency verifies that system efficiency can be improved. Figure 9b shows the relationship between the output voltage ripple and turn-on angle at different output power. Section 2 presents that the output voltage ripple is sensitive to turn-on angles. As shown in the figure, with the increase of the turn-on angle, the voltage ripple shows the trend of first decreasing and then increasing. So, there will be an optimized turn-on angle that minimizes output voltage ripple. Figure 9c shows the relationship between power converter loss and turn-on angle at different output power. Power electronics devices will have fewer on-time and switching times as the increment of turn-on angle, so the power converter loss is reduced. Combining the above concerned factors, Figure 9d shows the relationship between multi-objective evaluation results and turn-on angle, and that there will be an optimized turn-on angle for multi-objective optimization.

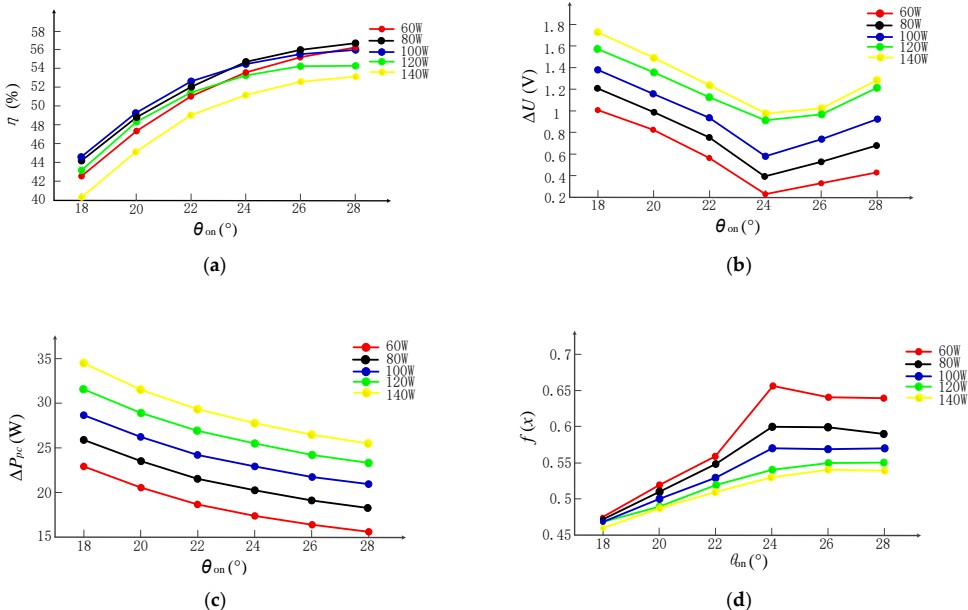

**Figure 9.** Relationship between concerned objectives and turn-on angle at different output power: (**a**) System efficiency versus turn-on angle at different output power; (**b**) Output voltage ripple versus turn-on angle at different output power; (**c**) Power converter loss versus turn-on angle at different output power; (**d**) Multi-objective evaluation versus turn-on angle at different output power.

Figure 10 shows the relationship between concerned control objectives and turn-on angle at different rotor speeds, where the turn-off angle is regulated according to (12) and the output power is fixed at 100 W. As shown in the figure, Figure 10 shows similar variation with Figure 9. Thus, the feasibility of proposed turn-on angle optimization strategy is demonstrated.

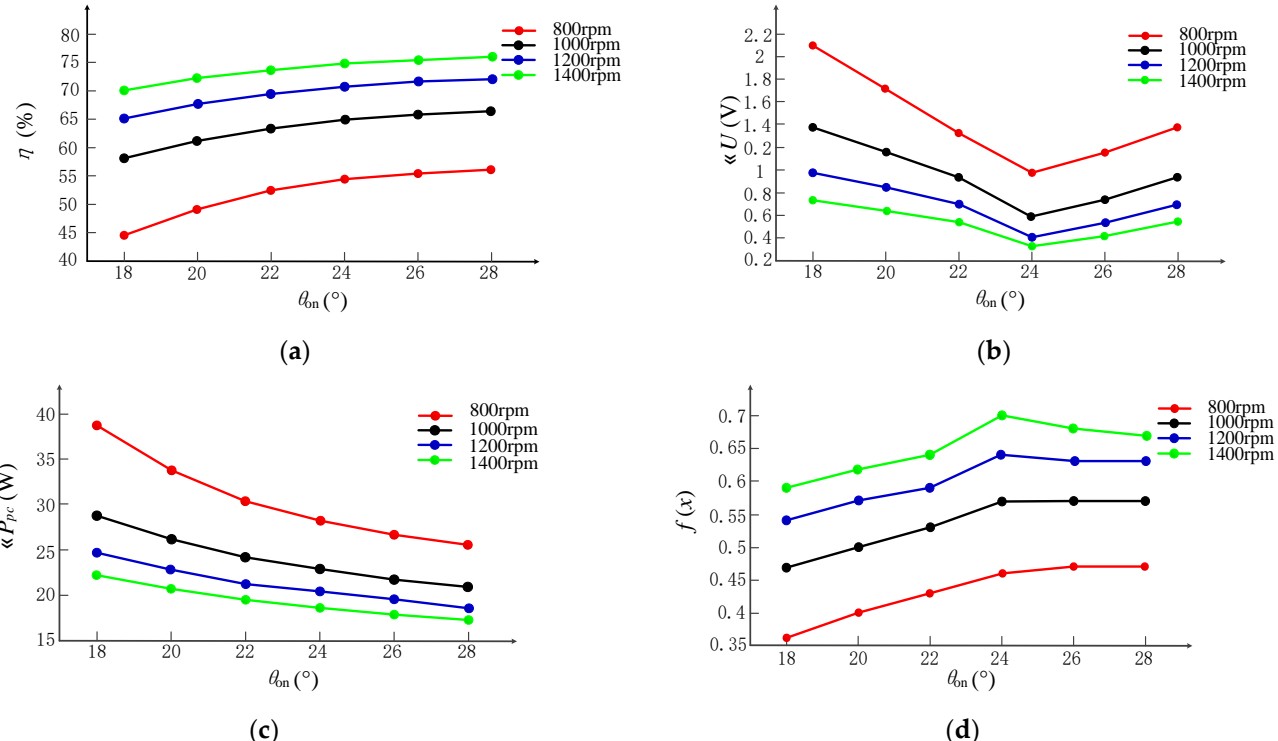

(**a**)

(**b**)

(**c**)

(**d**)

**Figure 10.** Concerned objectives versus turn-on angle at different rotor speeds: (**a**) System efficiency versus turn-on angle under different rotor speed; (**b**) Output voltage ripple versus turn-on angle at different rotor speeds; (**c**) Power converter loss versus turn-on angle at different rotor speeds; (**d**) Multi-objective evaluations versus turn-on angle at different rotor speeds.

## 5. Experimental Verification

In order to further validate the effectiveness of the proposed solution, the solution is also experimentally evaluated on the given SRG platform, the photograph of which is shown as Figure 11.

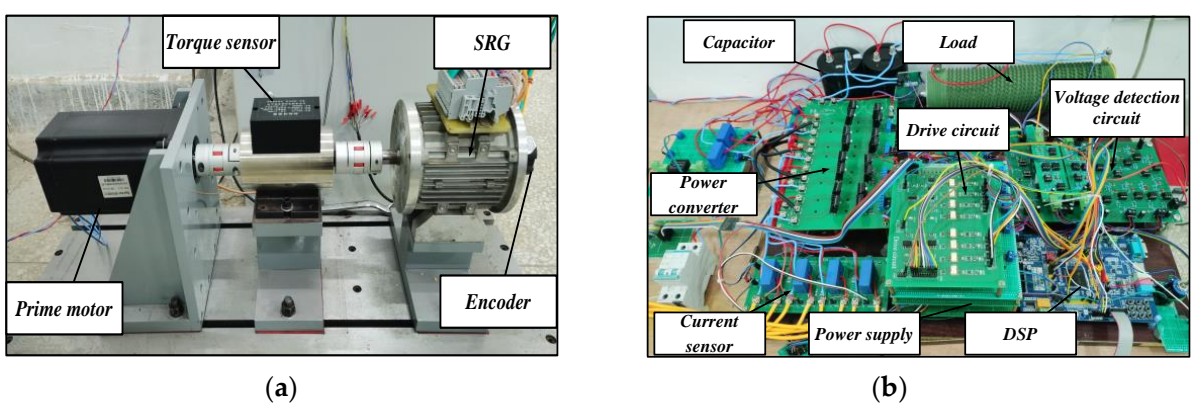

(**a**)

(**b**)

**Figure 11.** Photograph of the experimental platform: (**a**) Photograph of given SRG experimental platform; (**b**) Photograph of power converter.

Figure 12 shows the dynamic response for turn-on angle tuning at 800 rpm and the turn-off angle is regulated according to (12). The phase current at steady operation is shown at the top of the figure and SRG operates in the current control mode. The controller sets the output power as the top control objective. Figure 12a shows the dynamic response from 0 W to 50 W, from 50 W to 70 W and from 70 W to 50 W respectively. According to the experimental results, after the multi-objective optimization, the system efficiency is improved, the power converter loss is reduced and the voltage ripple is increased. Figure 12b shows the comparison of output voltage ripple before and after optimization. In order to improve the overall performance, voltage ripple is sacrificed to a certain extent. It can be seen from experiments that when the output power changes from 50 W to 70 W, the system efficiency is reduced, the voltage ripple is improved and the power converter loss is increased. Parameter changes eventually lead to a decrease in the value of the multi-objective function. When the output power changes from 70 W to 50 W, the change trend of the above parameters is reversed.

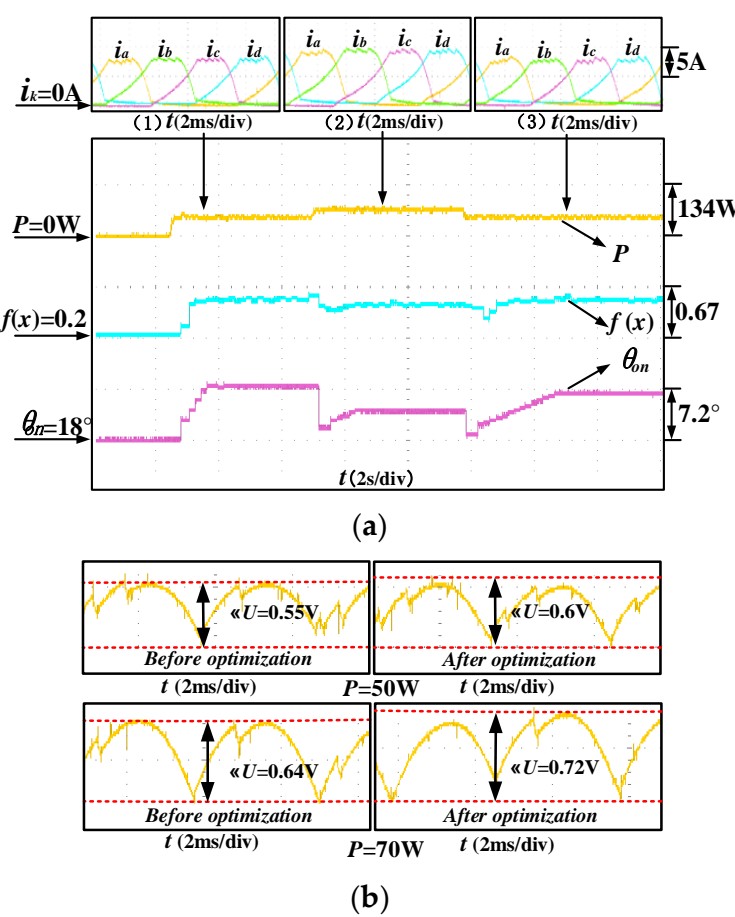

**Figure 12.** Dynamic response for turn-on angle tuning at 800 rpm.

Figure 13 shows the dynamic response for turn-on angle tuning at 1200 rpm and the turn-off angle is regulated according to (12). As shown at the top of the figure, SRG operates in single pulse mode. Figure 13a shows the dynamic response from 0 W to 100 W, from 100 W to 120 W and from 120 W to 100 W, respectively. According to the experimental results, after the multi-objective optimization, the system efficiency is improved, the power converter loss is reduced and the voltage ripple is increased. Figure 13b shows the comparison of voltage fluctuation before and after optimization, which is also sacrificed for improvement of overall performance.

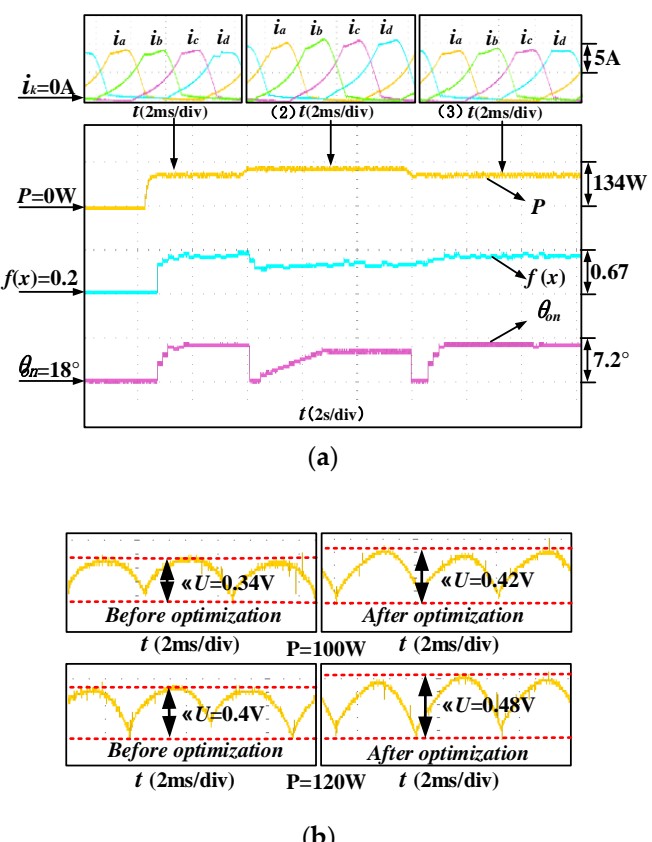

(a)

(b)

**Figure 13.** Dynamic response for turn-on angle tuning at 1200 rpm.

Figure 14 shows the output voltage response and output power response when load changes from 30 Ω to 15 Ω, from 15 Ω to 30 Ω, respectively, at 1200 rpm. The voltage closed-loop controller controls the output voltage to the target value by adjusting reference current. In Figure 14a, the target value of output voltage is set 20 V. When the resistance changes from 30 Ω to 15 Ω, the load increases. In order to stabilize the output voltage, the output power is increased. If the disturbance of output power exceeds the maximum output power variation range (5 W in Section 3), the turn-on angle will be re-optimized. Similarly, when the resistance changes from 15 Ω to 30 Ω, the load decreases, and the output power is decreased. Figure 14b set the target value of output voltage to 24 V and shows similar variation with Figure 14a.

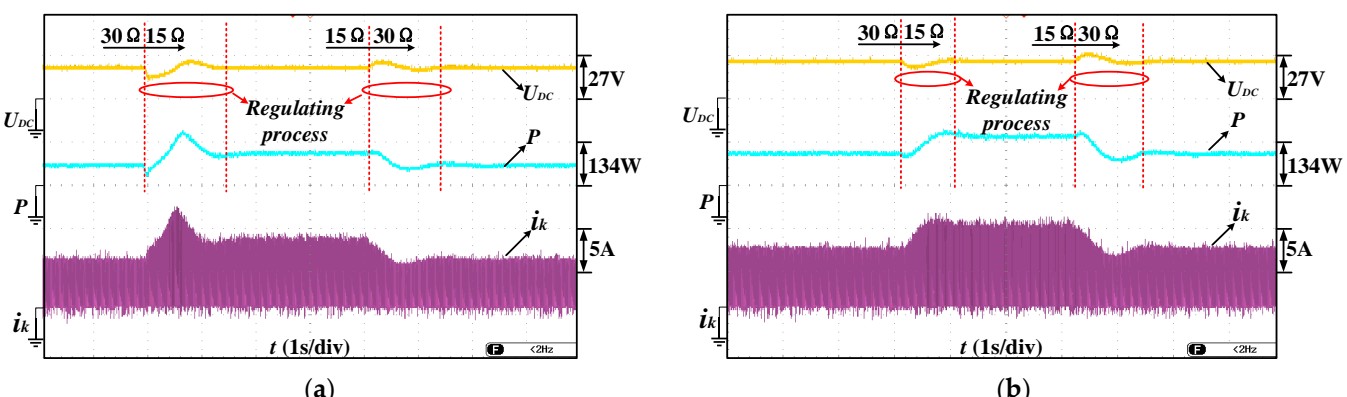

(a)

(b)

**Figure 14.** Output voltage response and output power response when load changes: (**a**) The target output voltage is set to 20 V; (**b**) The target output voltage is set to 24 V.

Figure 15 shows the dynamic response for turn-on angle tuning when load changes at 1200 rpm. When the resistance changes from 30 Ω to 15 Ω, the load increases. The disturbance of output power caused by the load change exceeds the maximum output power variation range, so the turn-on angle is re-optimized.

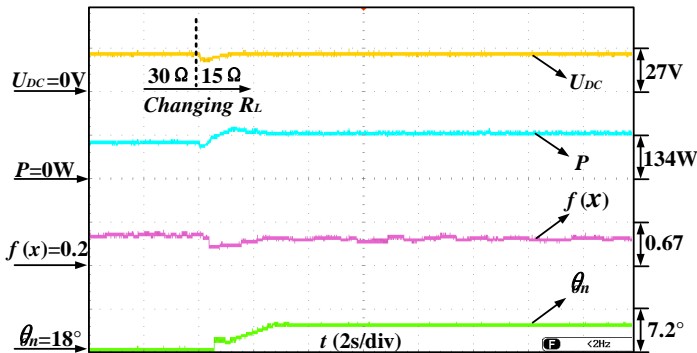

**Figure 15.** Dynamic response for turn-on angle tuning when load changes at 1200 rpm.

Figure 16 shows output voltage response and output power response when rotor speed changes. The output power in the Figure 16a is 70 W and the motor speed changes from 800 rpm to 1000 rpm, from 1000 rpm to 800 rpm, respectively. The output power in the Figure 16b is 120 W and the motor speed changes from 1000 rpm to 1200 rpm, from 1200 rpm to 1000 rpm, respectively. The power closed-loop controller remains the output power stable. However, changes in rotor speed can cause disturbances in the output power. If the disturbance exceeds the maximum output power variation range, the turn-on angle will be re-optimized.

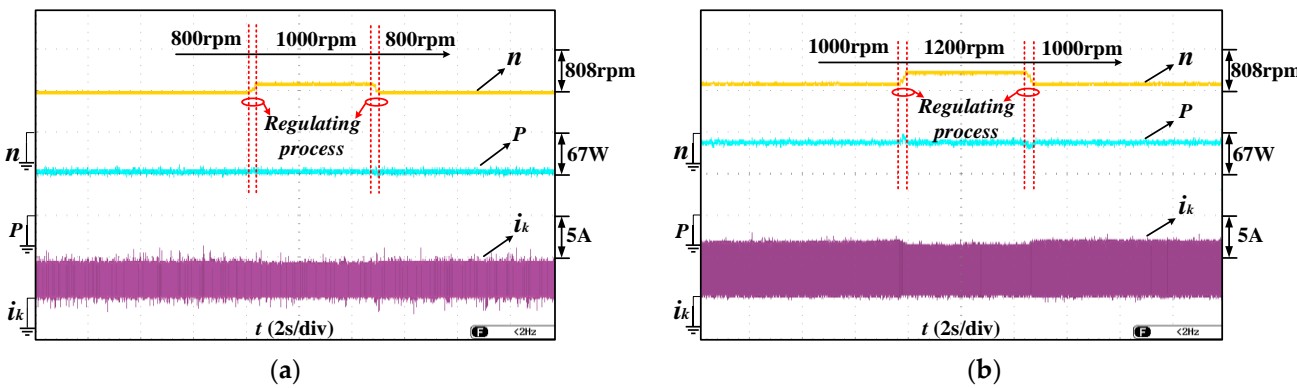

(**a**)　　　　　　　　　　　　　　(**b**)

**Figure 16.** Output voltage response and output power response when rotor speed changes: (**a**) The output power is 70 W and the motor speed changes from 800 rpm to 1000 rpm, from 1000 rpm to 800 rpm, respectively; (**b**) The output power is 120 W and the motor speed changes from 1000 rpm to 1200 rpm, from 1200 rpm to 1000 rpm, respectively.

Figure 17 shows the dynamic response for turn-on angle tuning on changing rotor speed and the output power is set to 150 W. When the rotor speed changes from 1000 rpm to 1200 rpm. There is a disturbance in the output power and the disturbance exceeds the maximum output power variation range. So, the turn-on angle is re-optimized.

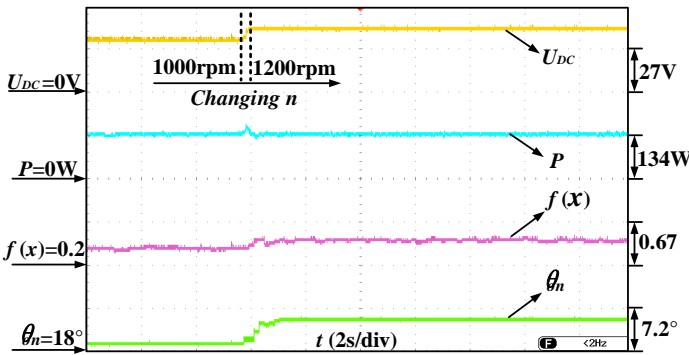

**Figure 17.** Dynamic response for turn-on angle tuning when rotor speed changes.

Figure 18 shows the effectiveness of the proposed solution. The output power is fixed at 50 W and 100 W respectively. As shown in Figure 18, after multi-objective optimization, system efficiency is improved and power converter loss is reduced. Although output voltage ripple is increased in concerned speed range, the overall performance is improved by the proposed solution.

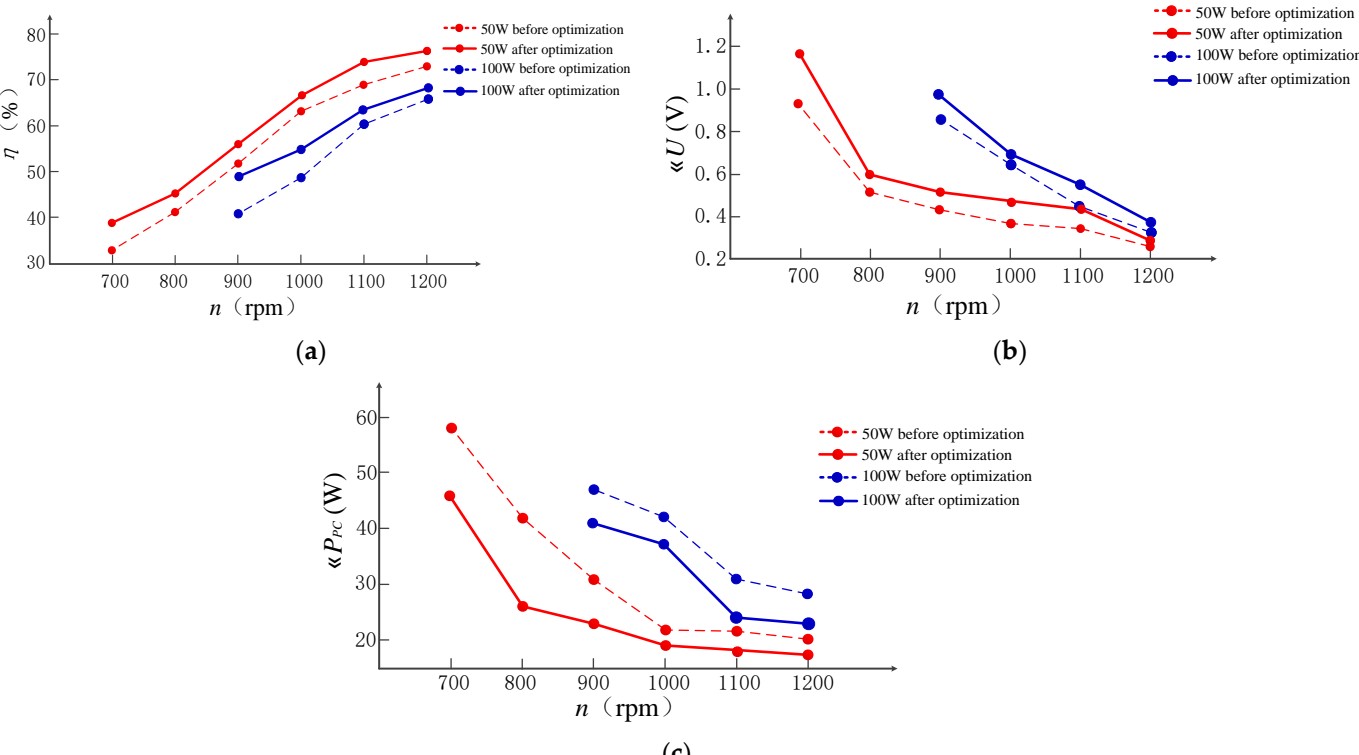

**Figure 18.** Effectiveness of proposed solution at different rotor speeds and output powers: (**a**) Comparison of system efficiency before and after optimization; (**b**) Comparison of output voltage ripple before and after optimization; (**c**) Comparison of power converter loss before and after optimization.

## 6. Conclusions

In this paper, in order to improve the overall performance of SRG, a multi-objective control strategy is proposed. The turn-off angle is optimized off-line to achieve maximum output power range. The turn-on angle is optimized on-line according to multi-objective evaluation results. The SAA is used to prevent falling into local optimum. The experimental results show that the output power can be guaranteed to reach the target value. After multi-objective optimization, system efficiency is improved, voltage ripple is increased and power converter loss is reduced. Although the voltage ripple is increased in the case

study, the overall performance of proposed system is improved. Besides, considering that the wind speed is inconstant, the maximum output power variation range is set. If the disturbance of output power caused by the change in wind speed exceeds the range, the turn-on angle is re-optimized. Additionally, the output power is set as the top control objective, and it can remain stable under different wind speeds. The proposed solution shows the following advantages:

(1) The controller can observe the system efficiency of SRG online with no need of complicated mathematical operations, and the proposed system has a low cost.

(2) The proposed solution can be flexibly applied to SRGs with different structures, which shows good reliability and is easy to implement.

(3) With the application of proposed solution, the overall operation performance of SRG can be improved, which avoids the disadvantage of single-objective optimization.

**Author Contributions:** Conceptualization, Q.W. and L.W.; methodology, Q.W., L.W., Z.J. and C.L.; software, Q.W., L.W., Z.J., W.X. and S.R.; validation, L.W., Z.J. and W.X.; formal analysis, Q.W., L.W., Z.J., S.R. and C.L.; investigation, S.R., W.X., J.D. and C.L.; resources, Q.W. and L.W.; data curation, L.W., W.X., Z.J. and J.D.; writing—original draft preparation, Q.W., L.W., W.X., S.R. and J.D.; writing—review and editing, L.W., Z.J. and W.X.; visualization, S.R., J.D. and C.L.; supervision, Q.W. and L.W.; project administration, Q.W. and L.W.; funding acquisition, Q.W. and L.W. All authors have read and agreed to the published version of the manuscript.

**Funding:** This research was funded by National Natural Science Foundation of China (Grant No. 51967013), Natural Science Foundation of Jiangxi Province (Grant No. 20212BAB214061) and Graduate Innovation Special Fund of Jiangxi Province (Grant No. YC2022—s131).

**Institutional Review Board Statement:** Not applicable.

**Informed Consent Statement:** Not applicable.

**Data Availability Statement:** Data available on request due to privacy restrictions.

**Conflicts of Interest:** The authors declare no conflict of interest.

## Nomenclature

| | |
|---|---|
| SRG, SRM | Switched reluctance generator, Switched reluctance machine |
| SAA | Simulated annealing algorithm |
| DC | Direct current |
| DPC | Direct power control |
| ASHB | Asymmetric half-bridge |
| $P$, $P_m$ | The output power, Absorbed mechanical power |
| $\Delta W$ | Mechanical energy |
| $U_{DC}$, $I_{DC}$ | DC bus voltage, Averaged output current |
| $U_k$, $i_k$ | Phase voltage, Phase current |
| $m$ | Phase number |
| $R_L$ | Load |
| $\theta_{on}$, $\theta_{off}$ | Turn-on angle, Turn-off angle |
| $\theta_{ext}$ | Angle positions where phase current extinguishes |
| $R_{ESR}$ | Equivalent resistance |
| $N_r$ | Rotor teeth number |
| $\omega$ | Angular velocity |
| $k$ | Phase A, Phase B, Phase C, or Phase D |
| $\varphi_x$ | The flux-linkage |
| $S_x$ | The area surrounded by flux-link trajectories |
| $P_{pc\text{-}in}$, $P_{pc\text{-}out}$ | The average input power of the converter, The average output power of the converter |
| $i_g$, $i_e$ | Total delivered current, Exciting current |
| $C$ | The output capacitance |

| $d$ | The chopping frequency for phase current control |
|---|---|
| $n$ | The actual rotor speed |
| $\eta$ | System efficiency |
| $\Delta U$ | Output voltage ripple |
| $\Delta P_{PC}$, $P^*$, $P(t)$ | Power converter loss, The required output power, The actual output voltage |
| $f(x)$ | Multi-objective evaluations |

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
