# Peer review of "Multi-Objective Control Strategy for Switched Reluctance Generators in Small-Scale Wind Power Generations"

_sustainability, doi:10.3390/su15076329_

Round 1

Reviewer 1 Report

 Comment in app

Reviewer 2 Report

Comments on

Multi-Objective Control Strategy for Switched Reluctance Generators in Small-Scale Wind Power Generations

The aim of the current study is to propose a multi-objective optimization control strategy to improve the static performance of SRG. In the proposed multi-objective control strategy, the turn-off angle is tuned to gain the maximum output power range at different rotor speeds; the reference current is regulated to achieve aim output power and the turn-on angle is tuned for multi-objective optimization.

The manuscript needs major revision. Furthermore, the following points must be addressed:

1- The abstract should be rewritten again in a scientific manner. It must be improved to present the objectives, the method, and the main findings from the research point in scientific way. Abstract, it needs to be enhanced by add the most important outcomes to show the amount of enhancement of the studied parameters.

2- what is these,  et. al. [5-7], in [13] etc..

3- The literature survey should be improved. The structure of the writing is not acceptable. It is not suitable for an academic paper.

4- When speaking about the previous work the tenses must in past.

5- Where is the nomenclature part.

6- There lots of typos error in the manuscript. So that, the English should be revised.

7- There are lots of long sentences which are not acceptable and are difficult to understand and followed.

8- The figure number 2 must be redrawn again. it is not clear. Use a different symbol.

9- Regarding the curve fitting in figure 5 what is the r2 value. There are more than point far from the fitted curve.  

10- The description of figure 6 should be improved. It is difficult for the reader to follow the description.

11- In figure 7, there is no explanation for the obtained results from the curve. Why at fixed rotor speed, the turn-off angle can be optimized for maximum output power.    

12- The discussions of the results, it must be improved with explain the reason for each outcome in the figures.

13- in figure 10 there is a load, is it fixed or variable load.

14- r/min     should be rpm this is the most common used unit.

15 - Conclusion part must be rewritten again. the reader could not know the beginning of the outcomes of the present work. It should be in points.

Round 2

Reviewer 2 Report

The authors response to all comments in the previous stage

the paper could be accepted after enhance figures 14, 15 and 16. The words and symbols inside the figures should be enhanced and its resolution and its font to be easy for the readers.
